# Effects of an Extract of the Brown Seaweed *Ascophylum nodosum* on Postprandial Glycaemic Control in Healthy Subjects: A Randomized Controlled Study

**DOI:** 10.3390/md21060337

**Published:** 2023-05-31

**Authors:** Aleksandra Konic Ristic, Sinead Ryan, Maha Attjioui, Shane O’Connell, Eileen R. Gibney

**Affiliations:** 1UCD School of Agriculture and Food Science, University College Dublin, D04V1W8 Belfiled, Dublin, Ireland; eileen.gibney@ucd.ie; 2UCD Institute of Food and Health, University College Dublin, D04V1W8 Belfield, Dublin, Ireland; 3Marigot Ltd., P43NN62 Carrigaline, Ireland; sinead.ryan@marigot.ie (S.R.); maha.attjioui@marigot.ie (M.A.); shane.oconnell@marigot.ie (S.O.); 4Shannon Applied Biotechnology Centre, Munster Technological University-Kerry, V92CX88 Tralee, Ireland

**Keywords:** brown seaweed, *Ascophyllum nodosum*, refined carbohydrates, postprandial glycaemia, insulin, diabetes, RCT, fucoidan, algal polyphenols, phlorotaninins

## Abstract

The effects of the consumption of an extract of the brown seaweed *Ascophyllum nodosum* (BSW) on postprandial glucose and insulin responses to white bread were investigated in an acute, randomized, double-blind, three-arm, crossover, controlled trial in healthy, normoglycemic subjects. Sixteen subjects were administered either control white bread (50 g total digestible carbohydrates) or white bread with 500 mg or 1000 mg of BSW extract. Biochemical parameters were measured in venous blood over 3 h. Significant inter-individual variation in the glycaemic response to white bread was observed. Analysis of the responses of all subjects to either 500 mg or 1000 mg of BSW extract versus control revealed no significant effects of treatments. The variation in response to the control was used to classify individuals into glycaemic responders and non-responders. In the sub-cohort of 10 subjects with peak glucose levels after white bread above 1 mmol/L, we observed a significant decrease in maximum levels of plasma glucose after the intervention meal with 1000 mg of extract compared with the control. No adverse effects were reported. Further work is warranted to define all factors that determine “responders” to the effects of brown seaweed extracts and identify the cohort that would benefit the most from their consumption.

## 1. Introduction

Numerous global challenges faced by humanity today are directly or indirectly related to food. Countries around the globe are facing major public health challenges due to the increasing prevalence of diet-related diseases. These conditions, such as obesity and diabetes, serve as major risk factors for cardiovascular diseases [1,2] and significantly contribute to mortality [3]. It is estimated that 463 million adults around the globe live with diabetes, and about 347 million have impaired glucose tolerance. The incidence of these conditions is projected to increase by up to 50% by 2045 [1]. At the same time, eating patterns and the food system in general are considered environmentally unsustainable [4], with food insecurity on the rise [5].

Starch-rich foods are a staple part of diets across the globe, representing an important source of macro- and micronutrients and contributing up to 80% of the total energy intake in some countries [6]. In terms of greenhouse gas emissions, energy consumption, land and water use, waste production, and effects on soil, they are considered more environmentally friendly than animal foods [7]. However, their effects on human health are not straightforward. Whole grains are recognized as a crucial component of a healthy and sustainable diet [7]. Their consumption has been linked to reduced rates of type 2 diabetes [8], CVD, and other chronic diseases [9]. However, the intake of refined grains and starchy vegetables is found to be associated with a higher risk of diabetes [9], major cardiovascular disease, stroke, and early death [10]. Furthermore, despite the recommended levels, the consumption of whole grains falls significantly short [11]. In Western countries, white bread, pastries, and other refined grain products remain a staple part of the diet, highlighting the need for an alternative approach to mitigate the adverse effects of high glycaemic index diets on health [12,13].

Food fortification and supplementation of the habitual diet with functional ingredients that have the potential to inhibit or slow down the digestion of carbohydrates and the uptake of simple sugars from the gut is a promising strategy for lowering the health risks of a starch-rich diet [14].

Algae are considered “blue foods”, with the lowest stress on the environment compared to other aquatic organisms [15]. Brown algae are rich sources of nutrients and bioactive compounds that have been shown previously to interact with carbohydrate metabolism [16]. The main bioactives in brown seaweed are specific polyphenols, phlorotannins, and bioactive sulfated polysaccharides [17]. The primary mechanisms of action for brown seaweed bioactives, which justify the need for additional human studies, involve the in vitro inhibition of enzymes involved in carbohydrate digestion, specifically salivary and pancreatic α-amylases and α-glucosidases [16,18,19,20]. Additionally, these bioactives exhibit the ability to inhibit trans-epithelial glucose transport [21]. 

Although brown seaweed extracts have shown potential for modulating carbohydrate metabolism in in vitro studies, their effects on biomarkers of postprandial glycaemic control in humans are limited and often inconsistent [22,23]. Several factors are potentially responsible for this inconsistency, including incomplete characterization of the brown seaweed extract [24] and variations in the composition of the extract across studies [22]. Additionally, the lack of information regarding the in vivo interaction of the main bioactives could also play a role in the observed inconsistencies. 

At the same time, glycaemic response to carbohydrates can vary significantly between individuals [25]. This variation is influenced by various factors, including the individual’s health status, age, sex [26], dietary patterns [27], level of physical activity [28], and ethnic background [24], among others. Another phenotype feature that has been shown to influence the glycaemic response to white bread and starchy foods is the level of the initial enzyme in starch digestion—salivary amylase [29]. As far as the authors are aware, the impact of salivary amylase levels on the potential of bioactive compounds to inhibit the digestion of starch has not been previously investigated. 

We have previously reported the comparative analysis of the chemical composition of several brown seaweed extracts from different algal species and the correlation with their potential to inhibit several enzymes of carbohydrate digestion [18]. The chemical composition of the *Ascophyllum nodosum* extract was analyzed, focusing on the distinctive composition of the fucoidan fraction. This analysis included assessing the size distribution of fucoidans and the specific ratio of polyphenols to fucoidans. These characteristics were found to be significant factors in the observed inhibition of alpha-glucosidases (specifically maltase and sucrase) in in vitro studies [18].

The present study aims to assess the effects of the brown seaweed *Ascophyllum nodosum* extract on postprandial glucose and insulin levels in response to white bread in normoglycaemic, healthy subjects. The role of variation in the glycaemic response to white bread was also addressed.

## 2. Results

### 2.1. Participant Characteristics

The baseline characteristics of participants are presented in Table 1. This study involved a total of sixteen participants, consisting of nine women and seven men, who completed the study. Values for all biochemical parameters were in the normal range, including biochemical parameters of cardiometabolic risk. Systolic blood pressure (SDB) and diastolic blood pressure (DBP) were also within normal values. Based on calculated body mass index (BMI) values, 4/16 participants were overweight (24.9–30 kg/m^2^) and one was obese (>30 kg/m^2^). 

Baseline salivary flow and the activity of α-amylase in saliva varied over a wide range of values among subjects, demonstrating considerable inter-individual variation. This variation, when combined, led to even more pronounced inter-individual variability in the secretion rate of salivary α-amylase. The resulting coefficients of variation were 55.98%, 70.58%, and 79.59% for salivary flow, salivary α-amylase activity, and salivary α-amylase secretion rate, respectively.

The distribution of fasting glucose levels on each of the three intervention days is shown in Appendix A. No differences in fasting glucose or insulin were found between the intervention days.

The distribution of fasting glucose values (Co) and maximal postprandial glucose values (Cmax) in response to control and intervention meals is shown in Figure 1. It is evident from the figure that there is a wide distribution of responses across the population group following each intervention. 

### 2.2. Effects on Postprandial Glucose Levels

No time × treatment interaction was found in the repeated measures of glucose levels (Figure 2). Peak plasma glucose concentrations (Cmax) were reached 30 min after the control meal and 45 min after low-dose brown seaweed meal (LDM) and high-dose brown seaweed meal (HDM) (Figure 2). Incremental Cmax values after the control, LDM, and HDM interventions were 1.7 (±1.06) mmol/L, 1.81 (±1.12) mmol/L, and 1.56 (±0.71) mmol/L, respectively, as mean (±SD), with no significant difference observed between them. 

No significant difference was observed at any time point in glucose concentrations or incremental areas under the curve (iAUC). iAUC values for glucose over 180 min after the control, LDM, and HDM interventions were 79.52 (±67.15) mmol × min/L, 116.4 (±19.8) mmol × min/L, and 81.25 (±63.44) mmol × min/L, respectively, as mean (±SD).

The peak glucose concentration (Cmax) for the control meal (white bread) for all 16 subjects was 6.78 (±1.11) mmol/L. Additionally, the incremental peak glucose concentration (iCmax) was 1.7 (±1.06) mmol/L as a mean (±SD), with a range of 0.28–3.52 mmol/L. Six out of sixteen subjects had an incremental glucose response to the control meal lower than 1 mmol/L. 

Based on the response to the carbohydrate load of the control meal and using 1 mmol/L as a reproducibly measurable increase for iCmax, we defined the “lower-normal” and “higher-normal” responder sub-cohorts, with the range in iCmax values between 0.28–0.97 mmol/L and 1.33–3.52 mmol/L, respectively. The values for parameters of postprandial glycaemia after the consumption of a control meal in two sub-cohorts are given in Table 2, with a significant difference between sub-cohorts observed for all of the parameters. Importantly, there was no observed difference in the fasting glucose levels between the two sub-cohorts.

In the “higher-normal” sub-cohort, we observed the time × treatment interaction in the repeated measures of glucose levels and a significant difference between both Cmax and iCmax values after HDM compared to the control meal (*p* < 0.05) (Figure 3). Incremental Cmax values after the control and HDM interventions were 2.38 (±0.69) mmol/L and 1.85 (±0.69) mmol/L, respectively. This indicates that the brown seaweed extract decreased the peak glucose concentration after the consumption of white bread by 22.3%. There was no effect of the intervention on iCmax in the lower-normal subcohort (Appendix A).

### 2.3. Effects on Postprandial Insulin Levels

No time × treatment interaction was found in the repeated measures of insulin level. No significant difference was observed at any time point in insulin concentrations or incremental areas under the insulin curve over 180 min after the control, LDM, or HDM interventions. Peak plasma insulin concentrations (Cmax) were reached at 45 min with all three intervention meals (Figure 4). Incremental peak insulin levels (iCmax) after the control, LDM, and HDM interventions as mean (±SD) were 29.48 (±14.14) mU/L, 26.8 (±13.45) mU/L, and 29.93 (±15.64) mU/L, respectively. Additionally, iAUC values over 3 h were 2201.61 (±986.54) mU × min/L, 1915.66 (±899.18) mU × min/L, and 2086.05 (±953.97) mU × min/L, respectively.

Insulin level, incremental insulin level, or iAUC for insulin after the consumption of a control meal between the two sub-cohorts were not significantly different. No time × treatment interactions were found in the repeated measures for any of the analyzed parameters either. 

No adverse or unexpected effects immediately following consumption, during the 24 h after the consumption, or anytime between two study visits were reported. 

## 3. Discussion

In this acute, randomized, crossover study, we investigated if the extract obtained from the brown seaweed *Ascophyllum nodosum* can modulate the rise in blood glucose and insulin levels observed in healthy subjects following consumption of a refined carbohydrate-rich meal. 

A significant variation in the glycaemic response to the control starch-rich meal (white bread) between subjects was observed in the trial. Compared to the control group, significant effects of interventions on either glucose or insulin levels were not observed in the total group of 16 subjects included in the trial. However, a shift in the time of the detected peak glucose values (tmax), from 30 min after the control meal to 45 min was observed after both intervention meals (with either a low or high dose of the extract). Clustering the subjects into two sub-cohorts based on their peak glucose response to the control meal allowed the comparison of responders with significantly different responses from baseline to the carbohydrate load in terms of iCmax, Cmax, or iAUC for glucose. In the sub-cohort of “higher-normal” responders, the brown seaweed extract significantly decreased the maximum incremental concentration of glucose after the starch-rich meal by 22% (*p* < 0.05). This supports the hypothesis that the apparent efficacy of the brown seaweed extract could depend on the individual’s glycaemic response to the carbohydrate load, with the extract primarily affecting subjects with higher spikes in glucose levels. 

Postprandial hyperglycaemia is a well-established, independent cardiovascular risk factor in patients with diabetes [30]. Moreover, epidemiological data indicate that it can be considered an independent predictor of cardiovascular events even in non-diabetic subjects [31]. The adverse effects of increased glucose levels following a meal are mediated by oxidative stress, directly proportional to glucose levels, which subsequently triggers an inflammatory response and impaired endothelial function and activates platelets and other atherothrombotic factors [31]. A significant decrease in flow-mediated vasodilatation, as a parameter of endothelial function, was observed in healthy subjects an hour after the consumption of a meal containing 40 g of carbohydrates [32]. The impairment of endothelial function induced by 75 g of glucose persisted even longer [32]. There is a hypothesis that atherosclerosis, the underlying cause of the vast majority of cardiovascular events, is a postprandial phenomenon [33]. It is evident that the risk caused by postprandial hyperglycaemia correlates with the level of hyperglycaemia itself rather than fasting glucose values. Consequently, the modulation of postprandial glycaemia in normoglycaemic individuals may be considered a rational approach to lowering the associated risk [34]. Our results have shown that subjects with higher postprandial glycaemia responded to the intervention with the brown seaweed *Ascophyllum nodosum* extract. 

In a recent study, Hall et al. used continuous glucose monitoring to evaluate the frequency of elevations in postprandial glucose over 2–4 weeks. They identified three “glucotypes”, i.e., glucose fluctuation patterns: low variability, moderate variability, and severe variability [35]. The evaluation of the glycemic responses to standardized meals highlighted the personal nature of glucose regulation and further confirmed the unfavorable effects of meals with refined carbohydrates. These types of meals were shown to cause a significant increase in glycaemia and a longer time in the pre-diabetic and diabetic range even in subjects with normal outcomes from standardized tests: fasting glucose, HbA1c, and postprandial response to glucose [35]. Our results rationalize further investigation of the effects of brown seaweed bioactives targeted specifically at the most vulnerable “glucotypes”.

The mechanisms that underlie the effects of brown seaweeds and their bioactives on carbohydrate metabolism and glucose homeostasis include: (a) inhibition of enzymes of carbohydrate digestion, salivary and pancreatic α-amylases and α-glucosidases (e.g., maltase glucoamylase and sucrase-isomaltase) [17,18]; (b) inhibition of glucose transport through the intestinal epithelium [21]; (c) stimulation of glucose uptake by skeletal muscles [36] and adipocytes [37,38]; (d) stimulation of insulin secretion from the pancreas directly [21] or indirectly via GLP-1 [39]; (e) increased sensitivity of peripheral tissues to insulin [36], and (f) inhibition of gluconeogenesis in the liver [40].

The extract tested in this trial was obtained through the process of biological activity-guided optimization based on the inhibition of enzymes of carbohydrate digestion, salivary α-amylase and α-glucosidases, as the main targets for its action. The main bioactives identified in the extract were polyphenols and sulfated polysaccharides [18]. The comparative analysis revealed that the balance between these two classes of bioactives is crucial for the biological activity of the extract against digestive enzymes [18]. A recent study reported the complexity of the polyphenol fraction of *Ascophyllum nodosum* extracts, with 12 identified polyphenols [41]. Further studies are warranted to evaluate the precise mechanisms of action of these compounds and their human metabolites and their interaction with other classes of bioactives against the targets relevant for glycaemic control.

The number of previous studies that tested the impact of brown seaweed extract on glycaemic control after a meal is limited. Paradis et al. [23] investigated the effect of a polyphenol-rich extract from *Ascophyllum nodosum* and *Fucus vesiculosus* on postprandial glycaemia/insulinemia induced by white bread. They reported the reduction in incremental area under the insulin curve and insulin sensitivity without the observed effects on glycaemia. The authors did not address the role of the response to the control meal, but the reported data show that mean peak glucose values for the total cohort were higher than in our cohort. In our study, there was no significant effect observed on insulin at any time point. While recognizing the significance and health advantages of enhanced insulin sensitivity, our primary focus was on mitigating postprandial hyperglycemia (defined as a primary outcome) due to its direct correlation with acute changes in biomarkers of cardiovascular disease (CVD) risk. In contrast, we did not observe a correlation with insulin levels [42]. Additionally, reducing postprandial glycaemia without increasing insulin levels is considered a beneficial postprandial effect as defined by the EFSA NDA Panel [43].

In another set of studies, the role of postprandial glycaemic response was examined by testing the effects of *Fucus vesiculosus* extract on glycaemia and insulinemia in response to white bread in the mornings and evenings. The studies reported lower responses to the challenge meal in the mornings than in the evenings [24,44]. However, there was no effect of the treatments in both settings on either postprandial glucose or insulin levels. The chemical analysis of several algal extracts on the market, including the one used in this trial, has revealed that the *Ascophyllum nodosum* extract has a unique composition concerning both polyphenols and poly/oligosaccharides [18]. Importantly, the mechanisms of enzyme inhibition were shown to be different, which could explain the difference in the observed in vivo effects compared to previous studies [18].

In a study on the acute effects of *Laminaria digitata* and *Undaria pinnatifida* seaweed on postprandial glycaemia in response to starch consumption, the authors reported the selective effects in participants weighing ≤ 63 kg [39]. This could be, at least partly, explained by the effect of body size on the glycaemic response standardized carbohydrate challenge, with subjects with a smaller body surface area showing higher responses to OGTT [45]. 

While no effect was seen on the total population, the current study identified a cohort of patients who were responsive to the treatments. In nutrition research, individuals’ responses to dietary interventions are known to be highly variable [25,46,47]. Identifying and comprehending these variations is crucial, as they can impact result interpretation and facilitate the development of customized nutrition advice and personalized nutrition delivery [48]. Tailoring nutrition or dietary advice to an individual based on their needs and requirements is known as precision nutrition and this approach can improve overall health as well as reduce the risk of diet-related diseases [49,50]. To support the rational design of future trials assessing the effects of brown seaweed bioactives on parameters of glycaemic control and reveal the full potential of brown seaweed for the individuals who can benefit the most from their consumption, it is crucial to clarify the influence of different subjects’ characteristics as biologically relevant factors for the observed effects. The reported difference in outcomes in two clusters of normo-glycaemic participants based on the peak glycaemic level supports the notion that the focus should be primarily on the factors that influence the response to the glycaemic challenge.

One of the factors that was previously reported to influence the postprandial glycaemic response to starchy foods is the activity of salivary α-amylase [29]. Salivary α-amylase is the enzyme involved in the first step in starch digestion and is encoded by the AMY1 gene. The activity of the enzyme is proportional to the diploid copy number (CN) for the AMY1 gene, which varies significantly among individuals [51]. In the study on healthy, lean individuals, AMY1 CN was shown not to be associated with the parameters relevant for glycaemic control, such as weight status, glucose tolerance, HOMA-IR or QUICKI insulin resistance, or Matsuda ISI [29]. However, AMY1 CN strongly correlated with the postprandial response to starchy foods, including potatoes, rice, and white bread [29]. Additionally, subjects with higher AMY1 CN and higher salivary α-amylase activity had 15–40% higher postprandial responses to starchy food compared to subjects with lower AMY1 CN [29]. Although the small sample size in our trial limits the validity of association analysis, the observed variation of α-amylase activity in saliva and other salivary parameters may influence the variation in the postprandial glycaemic response to the control and ultimately influence the effects of the intervention. Finally, the AMY1 CN and the associated salivary α-amylase activity, although widely unrecognized, are important sources of interindividual variation in postprandial response to starchy food and should be considered in designing relevant trials and evaluating tailored dietary interventions. 

Similar to our study, Wascher et al. [42] examined the effects of acarbose on flow-mediated vasodilatation (FMD) decrease in response to sucrose load in subjects with impaired glucose tolerance. The authors also performed a sub-analysis with respect to glycaemic responders to the challenge meal. In the study cohort, for the subjects above the median of the glycaemic response to sucrose, administering acarbose resulted in both a reduction in postprandial glycaemia and an improvement in FMD. However, in the group below the median, the response to sucrose was significantly lower, the FMD did not change, and there was no effect of acarbose on either postprandial glycaemia or FMD [42].

Synthetic alpha-glucosidase inhibitors (acarbose, voglibose, and miglitol) are therapeutics used for the treatment or prevention of type 2 diabetes and its complications [52]. They are effective in modulating postprandial hyperglycaemia and lowering glycated hemoglobin [53]. At the same time, they have a significant side effect profile, with abdominal symptoms (i.e., flatulence, abdominal fullness) reported in approximately 50% of patients [54]. Natural inhibitors of carbohydrate digestion enzymes with fewer side effects are widely promoted as a rational alternative [55], especially in healthy subjects and subjects at risk of developing diabetes, taking into account the risk level and an acceptable ratio between safety and efficacy [56]. 

There are some limitations to this study. Our primary outcome was iCmax; however, the participants were not recruited based on iCmax or any other parameter of glycaemic control. Secondly, the number of participants was calculated based on the results of a pilot study that reported a 50% higher iCmax value in response to a control meal. In addition, the study was not controlled for various factors reported to affect postprandial glycaemic response that might have contributed to the observed inter-individual variability. Nevertheless, we made efforts to minimize intra-individual variability by implementing controls at the individual level for certain factors such as diet, physical activity, and stress.

## 4. Materials and Methods

### 4.1. Study Design

This study was designed as an acute, randomized, double-blind, three-arm, crossover, controlled trial. The sample size was determined to enable the detection of a 0.5 mM difference in maximal incremental blood glucose levels (iCmax) between groups, assuming 80% power with an alpha of 0.05. It was based on the results from a pilot study in a similar cohort with iCmax values of 2.69 ± 0.79 mmol/L and 2.16 ± 0.56 mmol/L in the control group and the intervention group (500 mg BSW), respectively. The selected dose levels of the extract were: (1) 500 mg, which was the dose tested in the pilot study and previously reported in clinical trials on *Aschophyllum nodosum* extracts [23,24]; and (2) 1000 mg, which allowed the assessment of the dose–response relationship. This study was approved by the UCD Ethics Committee, and it was performed in accordance with the Declaration of Helsinki at the UCD Institute of Food and Health, Ireland. Participants provided informed consent before participating in any trial-related procedures and were assigned an anonymized study ID. This trial was registered at clinicaltrials.gov (NCT05460884). 

### 4.2. Participants

Participants were recruited through public advertisements (posters and leaflets) displayed around and in proximity to the campus. Inclusion criteria were as follows: healthy, i.e., with no known health problems; and age between 18–60 years. Exclusion criteria included smoking, pregnancy or lactation, previous diagnosis of any chronic illness (including diabetes, hypertension, gastrointestinal diseases, etc.), long-term prescribed medical therapy (except contraceptives), and allergy to fruits, vegetables, pollen, seafood, or seaweed. In addition, the study also excluded individuals who followed a specific diet or dietary regimen for weight management or engaged in regular consumption of fruit and herbal extract supplements. Additionally, individuals who had donated blood within four weeks prior to the study, intended to donate blood during the study, or planned to donate blood less than four weeks after the last study visit were also excluded. Participation in another trial involving dietary intervention or blood sampling was also considered an exclusion criterion. Participants were assessed for eligibility and recruited on an ongoing basis to complete the study with 16 subjects.

### 4.3. Study Protocol

Participants recruited for this study were scheduled for three study visits, at least seven days apart. The allocation sequence of three interventions was generated by block randomization (block size of six) using a computer random number generator (https://www.randomizer.org/, accessed on 18 September 2019)

Before the first study visit, participants were instructed to follow and record their usual diet and physical activity (PA) over the 3 days preceding the study. Participants were asked to follow the same dietary and PA regimen for 3 days before the next study visit and record their diet and PA again. On each study visit, all participants were instructed to fast for 12 h before their arrival, to freely consume water, and to have at least 7 h of sleep. After arriving at the site (the Human Intervention Suite at the UCD Institute for Food and Health) between 7.30 and 9 and after an initial 10 min rest, blood pressure, heart rate measurements, and anthropometric and body composition assessments were completed. Participants rested for 20 min before providing the first saliva, following which a nurse placed a cannula in an antecubital vein without taking any blood. A second sample of saliva was collected, followed by the baseline blood collection. Participants were then served the test meal (described below), and the time needed to complete consumption was recorded. All participants completed all test meals (100% compliance), and there were no dropouts during the study. Blood samples were collected again at 15, 30, 45, 60, 90, 120, 150, and 180 min after the first bite was reported to have been swallowed. Participants rested in a reclined position and remained sedentary for the duration of the study day. Each visit lasted ~4 h, and following completion, participants were provided with lunch. 

During each study visit, participants were given a list of common adverse and unexpected effects observed in clinical trials. They were then asked to report if they experienced any of them immediately following consumption, within 24 h after consumption, or anytime between two study visits. 

The graphical representation of the study design, the CONSORT 2010 checklist, and the CONSORT flow diagram are provided as Appendix A, respectively.

### 4.4. Anthropometric, Body Composition, Blood Pressure, and Heart Rate Measurements

Body height (cm) was measured using a stadiometer, and weight (kg), BMI, and body composition parameters (%fat, total fat, total visceral mass) were measured using a dual-frequency bioimpedance scale (Tanita, model DC430MA). Blood pressure and heart rate were measured using a digital blood pressure monitor (OMRON M6 Comfort, Omron Healthcare Co., Ltd., Kyoto, Japan). Blood pressure and heart rate were measured three times consecutively, and if the variation was less than 10%, the average value was calculated. 

### 4.5. Blood Collection and Biochemical Analyses

The first 2 mL of each blood sample drawn were discarded, followed by collection in three different tubes (Vacuette^®^, Greiner Bio-One, Kremsmünster, Austria): a NaF/K-Oxalate tube for the isolation of plasma (for glucose levels), a K2EDTA tube for plasma (for insulin levels), and clot activator tubes for serum. Tubes for plasma separation were kept on ice and centrifuged immediately at 1100× *g* for 10 min, aliquoted, and stored at −80 °C until further analysis. After at least 30 min at RT to allow clotting, serum was isolated by centrifugation (1100× *g*, 10 min, RT). Samples were aliquoted and stored at −80 °C until further analysis. 

Glucose in plasma and total triglycerides (TG), low-density lipoprotein cholesterol (LDLc), high-density lipoprotein cholesterol (HDLc), and total cholesterol (TC) in serum were measured using an enzymatic and colorimetric assay on a Pentra C400 Clinical Chemistry analyzer (HORIBA Medical, Montpellier, France). Insulin was determined by ELISA (Mercodia AB, Uppsala, Sweden). 

### 4.6. Saliva Collection and Analysis

Unstimulated total saliva was collected by the direct expectoration (spitting) method, as previously described [57]. Briefly, at each time point, subjects were seated comfortably and asked to allow spontaneous saliva flow for 1 min and expectorate the whole saliva content into previously weighed, sterile plastic tubes. This was repeated three times over 3 min, and samples were combined. Tubes were weighed and stored immediately at −80 °C for at least 4 h to allow mucin to precipitate, then thawed on ice. They were centrifuged for 15 min at 1500× *g* at 4 °C, separated from the sediment, and aliquoted. The salivary flow rate was calculated by dividing the sample volume (mL) by the time (min) taken to produce it. The volume of the collected saliva was determined by weighing, based on the average density of saliva of 1.00 g/m [58]. Amylase activity was determined using a commercial enzymatic assay (Salimetrics Europe Ltd., Newmarket, UK). Salivary amylase secretion rate (U/min) was calculated as amylase activity in the volume of saliva secreted per minute. 

### 4.7. Intervention and Control Meal

The tested product was a commercial, proprietary extract of the brown seaweed *Ascophyllum nodosum,* kindly provided by the company (Marigot Ltd., Carrigaline, Ireland). The extract was obtained through the multi-step process of biological activity-guided optimization, using the inhibitory potential against α-amylase and α-glucosidases as the assessed outcomes. The source and chemical characteristics of the extract used in the intervention and its effect on the enzymes of carbohydrate digestion were previously reported [18]. Briefly, the content of polyphenols, as a mean (±SD), was 6.57 (±0.33) *w*/*w*%, and the fucoidan content, calculated as the sum of fucose, sulfate, and other monosaccharides, was 19.34 (±0.14) *w*/*w*% [18]. The concentrations of the extract that induced 50% inhibition of maltase (10 mM) and sucrase (25 mM) activity (IC_50_ values) were 0.26 (±0.01) mg/mL and 0.83 (±0.21) mg/mL, respectively [18]. The extract was tested by Southern Scientific Services Ltd., a third-party laboratory, for its chemical and microbiological safety, and the intake of micronutrients present (i.e., iodine) with the intervention was confirmed to be in line with the recommendations [59]. Two different doses of the extract, 500 mg and 1000 mg per meal, were tested. Extracts were pre-measured in the test tubes labeled with the random intervention codes. Empty tubes were labeled with codes according to the allocation sequence indicated by the control intervention. For the duration of the study, all tubes were stored at −20 °C in a humidity protected environment. 

The control meal consisted of 108 g or approximately two white bread buns (Bundy’s originals, Johnston Mooney and O’Brien Bakery, Dublin, Ireland) that provided 50 g of available carbohydrates as measured by the commercial kit (Megazyme International Ireland Ltd., Bray, Ireland) and was always served fresh. 

The intervention meal consisted of the same amount of white bread (108 g or approximately 2 buns) and either 500 mg or 1000 mg of the brown seaweed extract distributed evenly between the two bun halves. Intervention and control meals were prepared daily according to the allocation sequence by technical staff not included in the trial and served in a double-blinded fashion (to participants and researchers). However, due to the taste of the extract, subjects were ultimately not considered blinded as they reported the taste revealed the higher dose intervention. 

### 4.8. Statistical Analysis

The change in the maximal incremental plasma glucose level (iCmax) between baseline and endpoint within intervention groups vs. control was predefined as the primary outcome measure of the study. Changes in insulin levels and the incremental area under the curve (iAUC) were secondary outcome measures. The area under curve (AUC) for glucose and insulin was calculated using the trapezoidal rule, and the net AUC was calculated by subtracting the AUC below baseline values. The normality of the obtained data was analyzed by the Shapiro–Wilk test. To analyze the glucose and insulin levels over time, a repeated measures ANOVA was conducted. Subjects were considered fixed effects, while treatment and time were treated as repeated measures. To account for multiple comparisons, a Bonferroni adjustment was applied. The correlation was evaluated using Pearson or Spearman correlation, as appropriate. Differences were considered significant at *p* < 0.05. Data in the text and tables are shown as the mean (±standard deviation; SD). In the figures, data are presented as mean (±standard error; SE). Statistical analyses were performed with SPSS software (version 27, IBM SPSS Statistics). GraphPad Prism (version 7.02; GraphPad Software, San Diego, CA, USA) was used for graphical representation. The sample size was determined based on the Cmax for glucose reduction as the primary outcome, obtained in a pilot study with a statistical power of 80%.

## 5. Conclusions

Our results demonstrate that 1000 mg of an *Ascophyllum nodosum* extract has the potential to modulate postprandial glycaemic response to a 50 g carbohydrate load from white bread. This effect was specifically shown in the sub-cohort of subjects with a sizeable glycaemic response (>1 mmol/L) to the challenge meal. This finding suggests that the 1000 mg treatment would be beneficial for subjects who may be at higher risk for postprandial hyperglycaemia. Additional investigation is warranted to explore the phenotype and/or genotype profiles that may define the consistent response to *Ascophyllum nodosum* bioactives and their effects on carbohydrate digestion and postprandial glycaemic control. *Ascophyllum nodosum* is an abundant source of nutrients and non-nutritive compounds, which, as part of blue food, has an important role in the shift towards healthy and sustainable food systems. However, strong evidence of its effect on health and a clear understanding of who may benefit the most from its consumption are essential for its use as a food or nutraceutical. 

## Figures and Tables

**Figure 1 marinedrugs-21-00337-f001:**
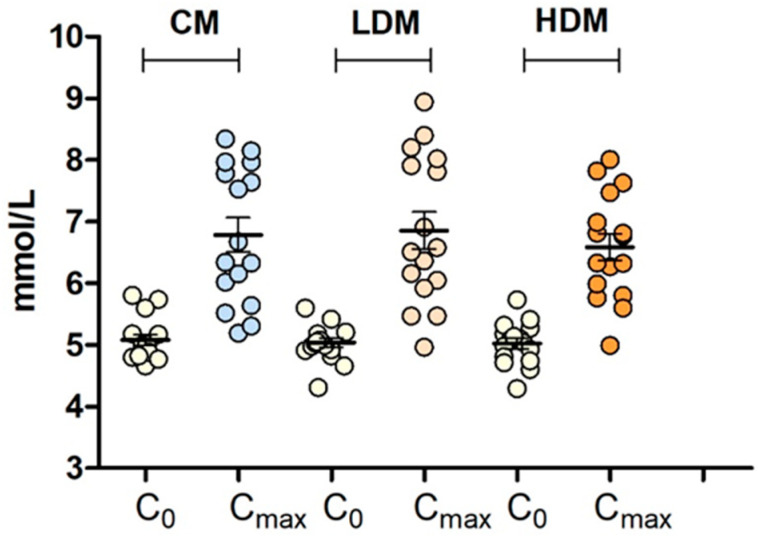
Scatter plot of fasting (Co) and maximum glucose levels (Cmax) after the control meal (CM), the intervention meal with a lower dose of brown seaweed extract (LDM), and the intervention meal with a higher dose of brown seaweed extract (HDM) in sixteen subjects. Values are means with standard errors.

**Figure 2 marinedrugs-21-00337-f002:**
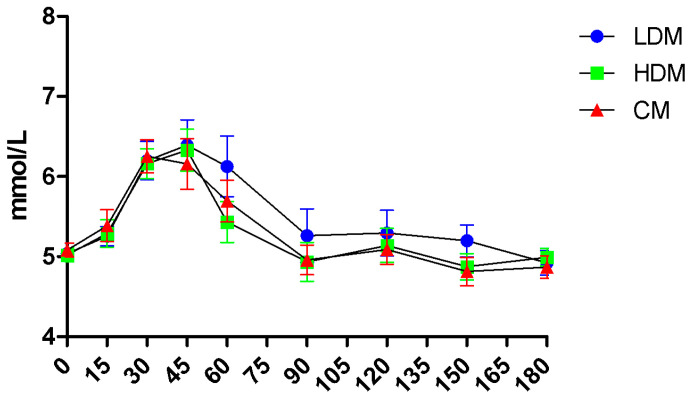
Concentrations of plasma glucose in sixteen healthy subjects over 3 h after the consumption of a control meal (CM), an intervention meal with a lower dose of brown seaweed extract (LDM), and an intervention meal with a higher dose of brown seaweed extract (HDM). Values are means with standard errors.

**Figure 3 marinedrugs-21-00337-f003:**
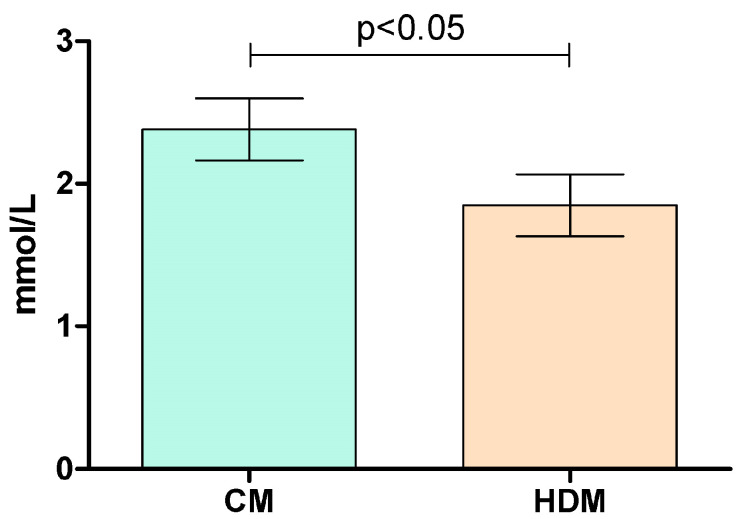
Incremental peak glucose concentration in a subcohort of 10 healthy subjects (“higher-normal” responders to refined starch load) after the consumption of a control meal (CM) and an intervention meal with a higher dose of brown seaweed extract (HDM). Values are means with standard errors.

**Figure 4 marinedrugs-21-00337-f004:**
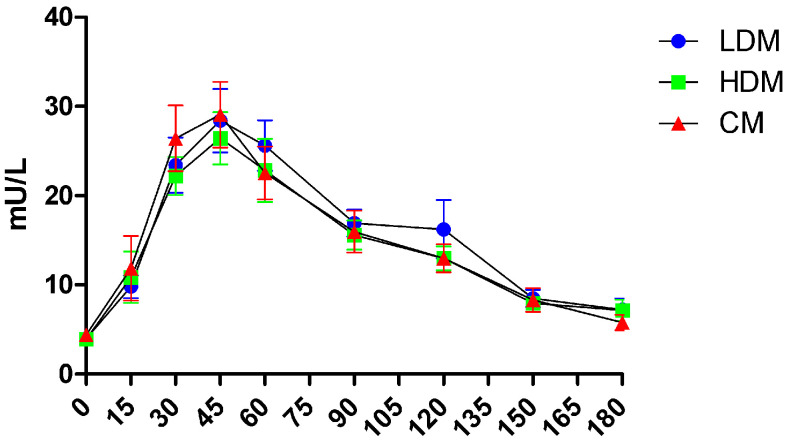
Concentrations of insulin in sixteen healthy subjects over 3 h after the consumption of a control meal (CM), an intervention meal with a higher dose of brown seaweed extract (HDM), and an intervention meal with a low dose of brown seaweed extract (LD). Values are means with standard errors.

**Table 1 marinedrugs-21-00337-t001:** Baseline clinical characteristics of the study participants.

Characteristics	Mean (SD)	Range
N	16	na
Male/Female	7/9	na
Age (years)	29.6 (±10.4)	18–57
Body weight (kg)	73.85 (±11.4)	60.5–93.4
BMI (kg/m^2^)	24.5 (±3.9)	18.1–32.1
DBP (mmHg)	110.4 (±9.2)	90–125
SDP (mmHg)	69.2 (±8.2)	52–82
Heart rate	68.6 (±11.5)	49–95
TG (mmol/L)	0.88 (±0.53)	0.39–2.26
TCHOL (mmol/L)	4.67 (±0.91)	3.47–6.31
HDL (mmol/L)	1.21 (±0.22)	0.79–1.49
LDL (mmol/L)	2.44 (±0.69)	1.29–3.49
Fasting plasma glucose (mmol/L)	5.11 (±0.32)	4.31–5.6
Salivary flow (mL/min)	0.393 (±0.22)	0.154–0.867
Salivary α-amylase activity (U/mL)	66.5 (±46.94)	16.63–181.53
Salivary α-amylase secretion rate (U/min)	37.73 (±30.03)	2.63–136.66

Data are presented as mean (±SD); BMI, body mass index; SDB, systolic blood pressure; DBP, diastolic blood pressure; TG, triglycerides; TCHOL, total cholesterol; HDL, high-density lipoprotein cholesterol; LDL, low-density lipoprotein cholesterol.

**Table 2 marinedrugs-21-00337-t002:** Parameters of postprandial glucose response to control in two sub-cohorts.

Parameters	“Lower-Normal”	“Higher-Normal”	Difference
Cmax	5.64 (±0.38)	7.47 (±0.75)	*p* < 0.0001
iCmax	0.57 (±0.29)	2.38 (±0.69)	*p* < 0.0001
iAUC180	19.83 (±25.72)	115.3 (±57.85)	*p* = 0.002

Data are presented as mean (±SD); Cmax, peak glucose concentration (mmol/L); iCmax, incremental peak glucose concentration (mmol/L); iAUC, incremental area under the glucose curve over 3 h (mmol × min/L).

## Data Availability

Access to study data is controlled. The data may be shared for further collaborative research projects in anonymized form after reviewing data access requests and following the “Good Practice Principles for Sharing Individual Participant Data from Publicly Funded Clinical Trials.” (Tudur Smith, C., Hopkins, C., Sydes, M., Woolfall, K., Clarke, M., Murray, G., Williamson, P., 2015; https://www.ukri.org/wp-content/uploads/2021/08/MRC-12082021-Datasharingguidance2015.pdf, assessed on 1 December 2019). Consent to share the data for such purposes is requested in the study consent form.

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
