# Peer review of "Effects of an Extract of the Brown Seaweed Ascophylum nodosum on Postprandial Glycaemic Control in Healthy Subjects: A Randomized Controlled Study"

_marinedrugs, 2023, doi:10.3390/md21060337_

Round 1

Reviewer 1 Report

The - small - study describes the effect of Ascolphyllum nodosum extracts on postprandial glucose and insulin response.
I have serious concerns about the acceptability of this article:
The sample size is extremely small - 16 subjects in total (7 men, 9 women), with very disparate ages and very different physiological characteristics. These variations would be acceptable if the sample size were much larger. Thus, it is impossible to establish any correlation with age, weight, BMI, etc.
Therefore, no correlation was made with the physiological status of the sample subjects. This would be important for understanding the impact of using the algae extract on at-risk groups, for example.
The chemical composition of the A. nodosum extract is not explained. This study would be the reason for the publication in the journal Marine Drugs. I do not understand the submission of this article to the journal, as it is clearly an article outside its scope.
The amount of data presented and the analysis performed is extremely simplistic. There are analyses, for example, of triglycerides and cholesterol, which allow us to understand other health parameters of the sample subjects, but this data is not used. 

I have nothing against the data itself, but I think the data should be supplemented with: 
1. analysis of the extracts and correlation with the chemical compounds in them, especially polyphenols which are known to have an effect on diabetes
2. a much larger sample size, to allow correlation with the physiological state of the individuals in the sample
3. if the goal is to improve glycemic control through the use of seaweed extracts, the metabolic mechanisms for this physiological response should be discussed.
Therefore, I don't think it has enough substance to be published in a scientific journal.

Reviewer 2 Report

Dear Authors,

This well designed RCT gives interesting insights into potential beneficial effects of seaweeds, that can help regulate glucose and insulin postprandial response. 

The work needs minor formal and language refinements  and explanation why samples of saliva was collected in relation to consumption of extracts of seaweed or glucose / insulin concentration measurements.

Please check yellow highlights with comments regarding the language style, grammar and clarity.

Reviewer 3 Report

General comment: Dear authors, many thanks for this well-done study. Please, find bellow a few minor comments for the manuscript improvement.

underpins the Introduction: Background well research

Results: Titles of graphical presentations should include the title of studied seaweed, Ascophylum nodosum.

Discussion:

It seems that new studies have been published about the effect of Ascophylum nodosum new, and the results of this study should be commented in comparison to the newest other studies.  

Results about Saliva collection and analysis should be commented within the discussion section

Reviewer 4 Report

This study determined the effects of brown seaweed Ascophyllum nodosum extract on postprandial glucose and insulin levels in response to white bread in normoglycaemic, healthy subjects.

Overall, it is well-written and informative. The following critique is offered for the authors’ consideration to improve this manuscript.

Line 372: the statement on the density of the saliva is 1.00 g/ml requires better referencing. Unsupported statement is not justified.

Section 4.7: what’s the rationale of choosing these 2 dosages, 500 mg and 1000 mg? Please explain why lower dosages, such as 250 mg was not used in this study.

Line 402-403: This statement is unclear/confusing. Reported post hoc?

Section 4.8: Explanation on sample size calculation requires further elaboration.

Table 2: Please explain why the SD value is bigger than the mean value for iAUC180 for ‘lower-normal’.

Why HbA1c was not determined in this study?

Another serious problem is that the manuscript simply reports scientific evidence, without highlighting the mechanisms of action that may be involved in the observed biological effect in relation to the phytochemical composition of this brown seaweed.

Round 2

Reviewer 4 Report

Authors have revised the manuscript accordingly. This manuscript can now be accepted for publication by Marine Drugs